# PMDformer: Patch-Mean Decoupling Information Transformer for Long-term Forecasting

**Ao Hu**[1,2*]  **Liangjian Wen**[1,7†]  **Jiang Duan**[1,6*]  **Yong Dai**[5‡]  **HE YAN**[6]  **Dongkai Wang**[1]
**Jun Wang**[1]  **Yukun Zhang**[4,2]  **Ruoxi Jiang**[2,3]  **Zenglin Xu**[2,3†]

[1]Southwestern University of Finance and Economics  [2]Shanghai Academy of AI for Science
[3]Fudan University  [4]Harbin Institute of Technology, Shenzhen  [5]X-Humanoid Research Institute
[6]Chengdu Everimaging Science and Technology Co., Ltd.
[7]Artificial Intelligence and Digital Finance Key Laboratory of Sichuan Province
{huao1105, wlj6816, zenglin}@gmail.com  duanj_t@swufe.edu.cn

## Abstract

Long-term time series forecasting (LTSF) plays a crucial role in fields such as energy management, finance, and traffic prediction. Transformer-based models have adopted patch-based strategies to capture long-range dependencies, but accurately modeling shape similarities across patches and variables remains challenging due to scale differences. To address this, we introduce patch-mean decoupling (PMD), which separates the trend and residual shape information by subtracting the mean of each patch, preserving the original structure and ensuring that the attention mechanism captures true shape similarities. Futhermore, to more effectively model long-range dependencies and capture cross-variable relationships, we propose Trend Restoration Attention (TRA) and Proximal Variable Attention (PVA). The former module reintegrates the decoupled trend from PMD while calculating attention output. And the latter focuses cross-variable attention on the most relevant, recent time segments to avoid overfitting on outdated correlations. Combining these components, we propose PMDformer, a model designed to effectively capture shape similarity in long-term forecasting scenarios. Extensive experiments indicate that PMDformer outperforms existing state-of-the-art methods in stability and accuracy across multiple LTSF benchmarks. The code is available at `https://github.com/aohu1105/PMDformer`.

## 1 Introduction

Long-term time series forecasting (LTSF) is a key task in machine learning, with wide applications in areas like energy management (Box & Jenkins, 1990), financial markets (Hu et al., 2025c), and traffic prediction (Guo et al., 2019; Yi et al., 2023b). Recent Transformer-based models have drawn inspiration from computer vision (Dosovitskiy et al., 2020), increasingly using patch-based strategies (Nie et al., 2023; Zhang & Yan, 2023; Chen et al., 2024; Wang et al., 2024c) to better capture long-range dependencies. Most of these approaches treat variables independently (VI) (Huang et al., 2025; Lin et al., 2024), while variable-dependent (VD) methods (Liu et al., 2024a; Luo & Wang, 2024) that model interactions across variables have not yet shown clear gains over VI baselines.

Unlike 2D images with a fixed spatial structure, time series are one-dimensional curve Germain et al. (2024); Hamilton (2020), with the primary focus being on capturing shape similarities between patches or variables (Grabocka et al., 2014; Kacprzyk et al., 2024) as well as modeling long-range trend (Li et al., 2023). For instance, two patches may share similar trends, such as gradual increases with comparable rates of change. Identifying such shape correspondence helps the model extract

---

[*]Equal contribution.
[†]Corresponding author.
[‡]Project leader.

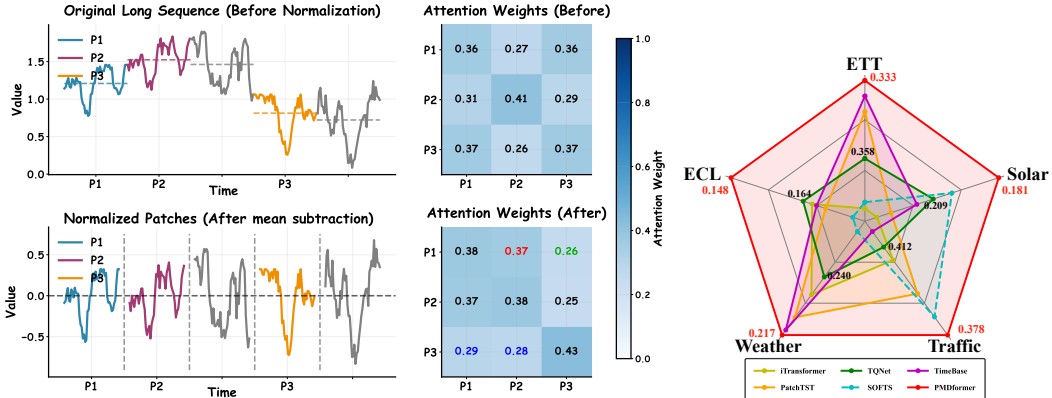

Figure 1: Attention weights for three patches before and after patch-mean decoupling. Scale differences initially obscure true shape similarity, which are clearly revealed after decoupling as increased (red) or decreased (green) correlations, with analogous similarity shown in blue for $(P_3, P_1)$ and $(P_3, P_2)$.

Figure 2: Comparison of the MSE of all baselines with our proposed PMDformer. The results are the averages for all prediction lengths.

temporally consistent patterns and improves forecast accuracy. However, time series data is inherently non-stationary (Fan et al., 2023; Liu et al., 2022b), where patch scales fluctuate wildly across time. As illustrated in the top panels of Figure 1, The attention weight of $(P_1, P_3)$ is higher than that of $(P_1, P_2)$, despite the more similar shape between $P_1$ and $P_2$. This occurs due to the different scales among $P_1$, $P_2$ and $P_3$, which influence the attention weights, thereby failing to reflect true shape similarity. Consequently, the model may learn incorrect similarity relationships, leading to performance degradation. Furthermore, this scale bias is even more pronounced when modeling dependencies between variables, further hindering the effectiveness of VD models.

To balance the scale differences of patches, recent methods have employed Patch Normalization (Liu et al., 2023b), which Z-score normalizes each patch by subtracting the mean and dividing by the standard deviation. However, the removal of the standard deviation inadvertently distorts the original shape of the patch. As a result, it hampers the model's ability to identify shape similarities across patches or variables. In this paper, we propose a simple yet effective alternative method called patch-mean decoupling (PMD). We subtract the mean of each patch, which recenters each patch to zero mean and explicitly separates the long-range trend component which is encoded in the means of patches from the residual shape information. Unlike Patch Normalization, our method preserves the original amplitude variations and maintains the intrinsic shape structure, ensuring that the model better captures true shape similarities across patches. As shown in Figure 1, through our method attention favors shape-aligned pairs $(P_1, P_2)$ over shape-unaligned $(P_1, P_3)$.

PMD thus enables more shape-focused attention across patches and variables, revealing true similarities obscured by scales. For cross-variable shape modeling, existing methods (Luo & Wang, 2024; Zhang & Yan, 2023) often compute interactions over the entire historical window. However, cross-variable relationships are often non-stationary and evolve over time, so recent interactions are the most predictive of future dynamics. For example, in financial markets asset correlations often spike sharply during crises. Relying on the entire historical dependencies introduces substantial noise and redundancy, degrading performance. To address this, we introduce proximal variable attention (PVA), which confines self-attention to the most recent patch—the segment most proximal to the prediction horizon. By capturing shape similarities among variables in this temporally relevant window, PVA minimizes noise from historical drifts and risk overfitting.

Complementarily, recentering via PMD inherently attenuates the long-term trend signal, potentially overlooking global dependencies. To restore this without disrupting shape matching between temporal patches, we propose trend restoration attention (TRA), which explicitly injects the decoupled means (long-range trend information) into the value pathway of the attention mechanism. This seamless integration allows the model to jointly encode local shape patterns and global trend yielding more stable forecasts.

Building on above, we propose **PMDformer**, which combines patch-mean decoupling (PMD) module, Proximal variable attention (PVA), trend reinsertion attention (TRA) and a projection layer for final forecasting. The comparison of predictive accuracy of our PMDformer and other state of the art models refer to Figure 2. Our contributions are:

- We introduce a novel mechanism to decouple trend and residual shape within the attention module via residual mean deduction, enabling more effectively capture shape similarity among temporal patches and varibles.

- We introduce proximal variable attention, which focuses on the most recent patch to capture the most relevant shape similarities, mitigating overfitting.

- We demonstrate the effectiveness of our approach through extensive experiments on a variety of LTSF benchmarks, showing that PMDformer provides more stable and accurate forecasts than current state-of-the-art methods.

## 2 RELATED WORK

Deep learning models have demonstrated remarkable performance in long-term time series forecasting. These models can be broadly divided into Transformer-based models Vaswani et al. (2017); Wu et al. (2021); Liu et al. (2022a); Zhou et al. (2022), MLP-based models Zeng et al. (2023); Li et al. (2023); Wang et al. (2024a); Hu et al. (2025b), GNN-based models Huang et al. (2023); Yi et al. (2023a) and CNN-based models Wang et al. (2023); Eldele et al. (2024); Hu et al. (2025a).

**Transformer-based time series models.** The success of Transformers (Vaswani et al., 2017) in NLP has inspired their adaptation for LTSF to capture long-range dependencies. Early models treat series as token sequences with efficient attention: Informer (Zhou et al., 2021) uses ProbSparse for complexity reduction; Pyraformer (Liu et al., 2022a) employs pyramidal attention; Autoformer (Wu et al., 2021) adds decomposition; and FEDformer (Zhou et al., 2022) incorporates frequency blocks. Yet, their efficacy is challenged by simple linear models (Zeng et al., 2023), underscoring needs for better temporal modeling.

**Patch-based time series models.** Inspired by vision transformers (Dosovitskiy et al., 2020), recent works segment time series into overlapping or non-overlapping patches to bolster local semantic capture. Transformer-based examples include PatchTST (Nie et al., 2023), which uses variable-independent shared encoders for temporal patch semantics (SOTA in LTSF), and Pathformer (Chen et al., 2024) with multi-scale patches and adaptive path selection for intra/inter-dependencies. MLP variants like TSMixer (Ekambaram et al., 2023) and PatchMixer (Gong et al., 2023) model patch relations via MLPs, while foundation models such as Moirai (Woo et al., 2024), Timer (Liu et al., 2024b), TimesFM (Das et al., 2024), and LLM-based (Pan et al., 2024; Jin et al., 2023)leverage patches for pretraining and cross-modal alignment. Recent TimeBase (Huang et al., 2025) employs orthogonalized patches to reduce redundancy for SOTA efficiency, which further underscores patches' success in LTSF modeling.

**Patch-Normalization.** Due to the non-stationary nature of time series, some works (Fan et al., 2023; Kim et al., 2021) apply normalization to mitigate scale discrepancies and stabilize distributions. Among them, Patch-level normalization works include SAN (Liu et al., 2023b), a model-agnostic framework that adaptively normalizes slices by removing non-stationarity for flexible forecasting, and SIN (Han et al., 2024b), which selectively learns normalization parameters to maximize local invariance and global variability, enabling interpretable long-term predictions. However, these normalization methods distort intrinsic patch shapes by scaling with standard deviation, hindering true shape similarity capture. In contrast, our PMD overcomes through mean subtraction to preserve amplitudes.

## 3 PROPOSED METHOD

We consider the task of long-term time series forecasting, where the goal is to predict the future evolution of multiple correlated variables given their historical observations. Formally, let $\mathbf{X} =$

$\{x_t \in \mathbb{R}^C \mid t = 1, 2, \ldots, L\}$ denote an input sequence of length $L$, where $C$ is the number of variables. Each $x_t = (x_t^1, x_t^2, \ldots, x_t^C)$ contains the values of all variables at time $t$. Given $\mathbf{X}$, the objective is to forecast the subsequent $T$ time steps $\hat{\mathbf{Y}} = \{\hat{x}_t \in \mathbb{R}^C \mid t = L + 1, \ldots, L + T\}$.

## 3.1 THE GENERAL STRUCTURE

Our proposed **PMDformer** architecture is a unified framework composed of four synergistic modules designed to explicitly decouple the long-term trend from the shape structure, selectively focus on the most relevant inter-variable dependencies, and ensure the accurate restoration of global dynamics for stable forecasting, as illustrated in Figure 3. (a) **Patch-Mean Decoupling (PMD)**: This module partitions the input time series into non-overlapping patches and explicitly separates each patch into its long-term trend component and its residual shape component. (b) **Proximal Variable Attention (PVA)**: To capture the most relevant cross-variable dependencies, the PVA module focuses its self-attention mechanism only on the $C$ tokens of the **last (proximal) patch**, modeling interactions across all variables. (c) **Trend Restoration Attention (TRA)**: This module is designed to model the shape similarities across patches. Crucially, it then **restores** the long-range trend information into the value pathway, enabling to accurately capture and utilize the overall long-term trend. (d) **Projection Layer**: This final layer combines the learned temporal representations with the reincorporated trend information through a fully connected projection to produce the final predictions.

## 3.2 MODEL ARCHITECTURE

**Patch-Mean Decoupling (PMD) & Embedding.** We first divide the input sequence $\mathbf{X} = \{x_t \in \mathbb{R}^C\}_{t=1}^L$ into $N$ non-overlapping patches of length $S$, where $N = \lfloor L/S \rfloor$. For variable $i \in [C]$ and patch index $j \in [N]$, the raw patch vector is

$$\mathbf{P}_j^i = \left( x_{(j-1)S+1}^i, x_{(j-1)S+2}^i, \ldots, x_{jS}^i \right) \in \mathbb{R}^S. \tag{1}$$

We then compute its temporal mean and the corresponding mean-decoupled residual:

$$\mu_j^i = \tfrac{1}{S} \sum_{k=1}^{S} x_{(j-1)S+k}^i, \qquad \mathbf{r}_j^i = \mathbf{P}_j^i - \mu_j^i \, \mathbf{1}_S, \tag{2}$$

where $\mathbf{1}_S$ is the $S$-dimensional all-ones vector. Each residual patch is then embedded into a $d$-dimensional representation through a shared linear projection. To encode location, we add learned positional embeddings to form the Transformer token:

$$\mathbf{P}_j^i := \mathbf{r}_j^i \mathbf{W}_E + \mathbf{b}_E + \mathbf{z}_{p_j} \tag{3}$$

where $\mathbf{W}_E \in \mathbb{R}^{S \times d}$, $\mathbf{b}_E \in \mathbb{R}^d$, and $\mathbf{z}_{p_j} \in \mathbb{R}^d$ denotes the positional embedding of patch $j$. By removing patch means before embedding, each patch is centered, which alleviates local inconsistencies across patches and variables so that attention mechanism can focus on shape similarities.

**Proximal Variable Attention (PVA).** Intuitively, accurate time series forecasting hinges on the immediate interactions between variables at the most recent time steps, as these dependencies are most indicative of near-term changes. Therefore, the PVA module is designed to concentrate its attention mechanism on the most **proximal** (i.e., most recent) tokens to model these critical cross-variable relationships.

Let $N$ be the index of the last (most recent) patch. We collect the most recent tokens of all $C$ variables, denoted as $\mathcal{P}_N = \{\mathbf{P}_N^1, \ldots, \mathbf{P}_N^C\}$, where each token $\mathbf{P}_N^i \in \mathbb{R}^d$ is derived from the Patch-Mean Decoupling (PMD) embedding. The PVA then applies Multi-Head Self-Attention (MHSA) exclusively within the set $\mathcal{P}_N$ to effectively capture the cross-variable shape dependencies that are most relevant for forecasting. Following the attention mechanism, a Feed-Forward Network (FFN) is employed to enhance the non-linear feature representation:

$$\hat{\mathcal{P}}_N = \mathrm{LayerNorm}(\mathrm{MHSA}(\mathcal{P}_N) + \mathcal{P}_N), \tag{4}$$

$$\mathcal{P}_N = \mathrm{LayerNorm}(\mathrm{FFN}(\hat{\mathcal{P}}_N) + \hat{\mathcal{P}}_N). \tag{5}$$

Tokens from the earlier historical patches, specifically those indexed $\{1, \ldots, N-1\}$, maintain their original representation derived from the PMD module. Following the PVA operation, the refined

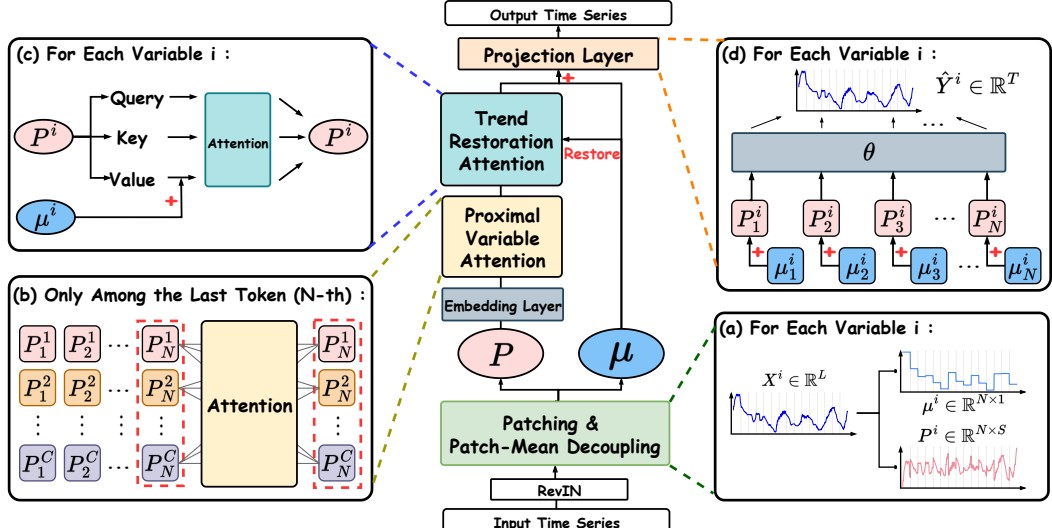

Figure 3: Overview of the proposed PMDformer. The model comprises: (a) **Patch-Mean Decoupling** module re-centers each patch and separates patches into trend and shape components; (b) **Proximal Variable Attention** operates only on the most recent token to capture variable interactions which are most relevant for forecasting; (c) **Trend Restoration Attention** restores long-range trends after value projections, restoring trend modeling; (d) **Projection Linear** adds the trend back to model long-range trend information for stable and accurate predictions.

token set $\mathcal{P}_N$ is concatenated with these remaining historical tokens along the patch dimension to form the full sequence of shape embeddings, denoted as $\mathcal{P} \in \mathbb{R}^{C \times N \times d}$. This deliberate strategy of restricting cross-variable attention solely to the most proximal patch offers dual advantages: it **enhances model robustness** by avoiding spurious long-range couplings from historical noise, and it improves **computational efficiency** by reducing the complexity from $O(C^2 N)$ to $O(C^2)$.

**Trend Restoration Attention (TRA).** Following the refinement of the most proximal tokens by the PVA module, the TRA module aims to capture temporal shape similarities across all historical patches while preserving long-range trend information. This is achieved by applying a parameter-shared Transformer encoder (MHSA + FFN) along the patch axis for each variable independently.

In this design, the Query($\mathbf{Q}$) and Key($\mathbf{K}$) projections operate solely on the shape embeddings, ensuring that the resulting attention scores $\mathcal{A}$ emphasize precise inter-patch shape similarity. To counteract the potential loss of global dynamics inherent in shape-focused modeling, we explicitly **incorporate the per-patch mean** ($\mu^i$) into the **Value** ($\mathbf{V}$) pathway. The additive reintegration is inspired by residual connections in ResNet (He et al., 2016). Concretely, for the $i$-th variable's patch sequence $\mathbf{P}^i \in \mathbb{R}^{N \times d}$, the computation is defined as:

$$\mathbf{Q}^i = \mathbf{P}^i \mathbf{W}_Q, \qquad \mathbf{K}^i = \mathbf{P}^i \mathbf{W}_K, \tag{6}$$

$$\mathcal{A} = \text{Softmax}\left(\frac{\mathbf{Q}^i (\mathbf{K}^i)^\top}{\sqrt{d}}\right), \tag{7}$$

$$\mathbf{V}^i = \mathbf{P}^i \mathbf{W}_V + \mu^i, \tag{8}$$

where $\mathbf{W}_Q, \mathbf{W}_K, \mathbf{W}_V$ are the projection matrices, and $\mu^i$ is the per-patch mean (Eq. 2), broadcast to match the dimensions of $\mathbf{P}^i \mathbf{W}_V$. This architectural separation allows the $\mathbf{Q}/\mathbf{K}$ pathway to model fine-grained local shape dependencies, while the $\mathbf{V}$ pathway ensures the preservation of the essential **global trend dynamics**. The resulting trend-integrated tokens are then refined through a Feed-Forward Network (FFN) to enhance the temporal representation learning.

**Projection Layer.** The temporal tokens produced by the TRA module are rich in shape dependencies but still require the final **restoration of the global trend information** for stable and accurate multi-step forecasting. This final step is essential to fully recover the original scale and long-term

dynamics that were decoupled earlier. To achieve this, before generating the multi-step forecasts, we **re-incorporate** the per-patch trend means ($\mu^i$) into the refined shape embeddings:

$$\hat{\mathbf{Y}}^i = (\mathbf{P}^i + \mu^i)\,\mathbf{W}_o + \mathbf{b}_o, \quad \hat{\mathbf{Y}}^i \in \mathbb{R}^T. \tag{9}$$

Here, $\mathbf{W}_o \in \mathbb{R}^{(N \times d) \times T}$ and $\mathbf{b}_o \in \mathbb{R}^T$ are the weight matrix and bias vector, respectively. The mean $\mu^i$ is implicitly broadcast to align with the dimensions of $\mathbf{P}^i$. This final step ensures the model's predictions are well-calibrated with the long-range trend observed in the input series.

### 3.3 THEORETICAL ANALYSIS

**Scale Bias Without Patch-Mean Decoupling (PMD)** Consider embedding raw patches $\tilde{\mathbf{x}} = \mathbf{r} + \mu\mathbf{1}$, where $\mathbf{r}$ is the residual and $\mu$ is the patch mean. The attention logit between tokens $(i, j)$ is given by:

$$\tilde{z}_{ij} = \mathbf{q}_i^\top \mathbf{k}_j = \tilde{\mathbf{x}}_i^\top \mathbf{M}\tilde{\mathbf{x}}_j = \underbrace{\mu_i\mu_j\mathbf{1}^\top\mathbf{M}\mathbf{1}}_{\text{mean–mean}} + \underbrace{\mu_i\mathbf{1}^\top\mathbf{M}\mathbf{r}_j + \mu_j\mathbf{r}_i^\top\mathbf{M}\mathbf{1}}_{\text{cross}} + \underbrace{\mathbf{r}_i^\top\mathbf{M}\mathbf{r}_j}_{\text{residual similarity}}, \tag{10}$$

where $\mathbf{M} := \mathbf{W}_E^\top\mathbf{W}_Q^\top\mathbf{W}_K\mathbf{W}_E$ and $\mathbf{1}$ is the all-ones vector. The first three terms depend on the means and can affect even dominate the residual similarity, inducing scale bias.

**Proposition 1: Sufficient Condition for Level-Dominated Logits** Let $i$ be a fixed query. A sufficient condition for the mean-dependent part of $\tilde{z}_{ij}$ to dominate the residual similarity uniformly over all keys $j$ is:

$$|\mu_i||\mu_j||\mathbf{1}^\top\mathbf{M}\mathbf{1}| \ge \|\mathbf{M}\|_2\|\mathbf{r}_i\|\|\mathbf{r}_j\| + |\mu_i|\|\mathbf{M}\mathbf{1}\|\|\mathbf{r}_j\| + |\mu_j|\|\mathbf{M}\mathbf{1}\|\|\mathbf{r}_i\|, \tag{11}$$

where $\|\cdot\|_2$ represents the spectral norm. This condition guarantees that the mean-dependent terms outweigh the residual term and cross terms, leading to scale-induced bias in attention. This confirms that attention is biased toward scale when the means are large, which motivates the need for patch-mean decoupling in our method.

## 4 EXPERIMENT

### 4.1 EXPERIMENT SETUP

**Datasets** We conduct experiments on 8 widely-used and publicly available real-world datasets. These include: ECL[1], Traffic[2], Weather[3], Solar[4], ETTh1, ETTh2, ETTm1, and ETTm2[5]. Following the experimental protocol established in prior work (Wang et al., 2024b; Qiu et al., 2024; Liu et al., 2023a), we partition the datasets into training, validation, and test sets with the following ratios: 6:2:2 for the four ETT datasets and 7:1:2 for the remaining datasets. The detailed statistics of each dataset are summarized in Table 1.

**Baselines** We compare PMDformer against 9 baselines, including state-of-the-art (SOTA) long-term forecasting models: TQNet (Lin et al., 2025), TimeBase (Huang et al., 2025), SOFTS (Han et al., 2024a), SparseTSF (Lin et al., 2024), ModernTCN (Luo & Wang, 2024), iTransformer (Liu et al., 2024a), TimeMixer (Wang et al., 2024a), and PatchTST (Nie et al., 2023).

**Setups** Consistent with prior research (Huang et al., 2025), we use an input length $L$ of 720 and evaluate prediction lengths $T$ of $\{96, 192, 336, 720\}$. Results for TimeBase, SparseTSF, iTransformer, TimeMixer, and PatchTST are derived from the TimeBase study, while other outcomes are from our own experiments. All experiments are conducted using PyTorch (Paszke et al., 2019) on an NVIDIA A100 80GB GPU. The Adam optimizer (Kingma, 2014) is employed, with learning rates chosen from $\{2e\text{-}4, 5e\text{-}4, 1e\text{-}3, 1e\text{-}2\}$. The number of patches $N$ is adjusted based on the requirements of each dataset.

---

[1]https://archive.ics.uci.edu/ml/datasets/ElectricityLoadDiagrams20112014

[2]https://pems.dot.ca.gov/

[3]https://www.bgc-jena.mpg.de/wetter/

[4]http://www.nrel.gov/grid/solar-power-data.html

[5]https://github.com/zhouhaoyi/ETDataset

Table 1: Characteristics of Long-term Time Series Datasets. This table summarizes key attributes of each dataset, including the application domain; the number of variables; total time points; data split ratios for training, validation, and testing and sampling interval.

| Domain | Electricity | | | | | Weather | Energy | Transportation |
|---|---|---|---|---|---|---|---|---|
| Dataset | ETTh1 | ETTh2 | ETTm1 | ETTm2 | ECL | Weather | Solar-Energy | Traffic |
| Variables | 7 | 7 | 7 | 7 | 321 | 21 | 137 | 862 |
| Time Points | 14,400 | 14,400 | 57,600 | 57,600 | 26,304 | 52,696 | 52,560 | 17,544 |
| Split Ratio | 6:2:2 | 6:2:2 | 6:2:2 | 6:2:2 | 7:1:2 | 7:1:2 | 7:1:2 | 7:1:2 |
| Sampling | 1 hr | 1 hr | 15 min | 15 min | 1 hr | 10 min | 10 min | 1 hr |

Table 2: Comprehensive results for multivariable time series forecasting with a lookback window of 720 time steps. Performance metrics for TQNet (Lin et al., 2025) and SOFTS (Han et al., 2024a) were obtained through our experiments, while results for other methods were sourced from Time-Base (Huang et al., 2025). The best results are highlighted in **bold**, and the second-best are indicated with underlining.

| | | PMDformer (ours) | | TQNet (2025) | | TimeBase (2025) | | SOFTS (2024) | | SparseTSF (2024) | | ModernTCN (2024) | | iTransformer (2024) | | TimeMixer (2024) | | PatchTST (2023) | |
|---|---|---|---|---|---|---|---|---|---|---|---|---|---|---|---|---|---|---|---|---|
| | Metric | MSE | MAE | MSE | MAE | MSE | MAE | MSE | MAE | MSE | MAE | MSE | MAE | MSE | MAE | MSE | MAE | MSE | MAE |
| ECL | 96 | **0.122** | **0.214** | 0.143 | 0.244 | 0.139 | 0.231 | 0.133 | 0.229 | 0.139 | 0.239 | 0.131 | 0.227 | 0.135 | 0.233 | 0.142 | 0.247 | 0.141 | 0.240 |
| | 192 | **0.140** | **0.231** | 0.151 | 0.247 | 0.153 | 0.255 | 0.160 | 0.255 | 0.155 | 0.250 | 0.145 | 0.241 | 0.155 | 0.253 | 0.159 | 0.256 | 0.156 | 0.256 |
| | 336 | **0.152** | **0.245** | 0.166 | 0.261 | 0.169 | 0.262 | 0.182 | 0.277 | 0.171 | 0.265 | 0.162 | 0.261 | 0.169 | 0.267 | 0.169 | 0.270 | 0.172 | 0.267 |
| | 720 | **0.177** | **0.272** | 0.194 | 0.291 | 0.207 | 0.294 | 0.224 | 0.310 | 0.208 | 0.300 | 0.193 | 0.289 | 0.204 | 0.301 | 0.209 | 0.313 | 0.208 | 0.299 |
| | Avg | **0.148** | **0.241** | 0.164 | 0.261 | 0.167 | 0.258 | 0.175 | 0.268 | 0.180 | 0.264 | 0.158 | 0.255 | 0.166 | 0.264 | 0.170 | 0.272 | 0.169 | 0.266 |
| Traffic | 96 | **0.338** | **0.212** | 0.398 | 0.297 | 0.394 | 0.267 | 0.355 | 0.255 | 0.389 | 0.268 | 0.382 | 0.267 | 0.374 | 0.273 | 0.396 | 0.294 | 0.363 | 0.250 |
| | 192 | 0.367 | **0.227** | 0.397 | 0.277 | 0.403 | 0.271 | **0.365** | 0.258 | 0.399 | 0.272 | 0.393 | 0.271 | 0.393 | 0.283 | 0.404 | 0.295 | 0.382 | 0.258 |
| | 336 | **0.379** | **0.235** | 0.403 | 0.279 | 0.417 | 0.278 | 0.390 | 0.278 | 0.417 | 0.279 | 0.409 | 0.277 | 0.409 | 0.292 | 0.419 | 0.302 | 0.399 | 0.268 |
| | 720 | **0.426** | **0.262** | 0.448 | 0.304 | 0.456 | 0.298 | 0.429 | 0.294 | 0.449 | 0.299 | 0.452 | 0.305 | 0.450 | 0.314 | 0.458 | 0.309 | 0.432 | 0.289 |
| | Avg | **0.378** | **0.234** | 0.412 | 0.289 | 0.418 | 0.279 | 0.385 | 0.271 | 0.414 | 0.280 | 0.408 | 0.280 | 0.407 | 0.291 | 0.419 | 0.300 | 0.394 | 0.266 |
| Weather | 96 | **0.141** | **0.181** | 0.160 | 0.213 | 0.146 | 0.198 | 0.165 | 0.219 | 0.174 | 0.231 | 0.155 | 0.210 | 0.159 | 0.212 | 0.163 | 0.223 | 0.149 | 0.199 |
| | 192 | **0.185** | **0.226** | 0.212 | 0.261 | 0.185 | 0.241 | 0.213 | 0.258 | 0.216 | 0.267 | 0.205 | 0.256 | 0.203 | 0.252 | 0.201 | 0.254 | 0.193 | 0.243 |
| | 336 | **0.236** | **0.274** | 0.260 | 0.299 | 0.236 | 0.281 | 0.272 | 0.305 | 0.260 | 0.299 | 0.255 | 0.290 | 0.253 | 0.291 | 0.258 | 0.300 | 0.240 | 0.281 |
| | 720 | **0.305** | **0.323** | 0.328 | 0.343 | 0.309 | 0.331 | 0.380 | 0.371 | 0.325 | 0.345 | 0.317 | 0.336 | 0.317 | 0.337 | 0.329 | 0.348 | 0.312 | 0.334 |
| | Avg | **0.217** | **0.251** | 0.240 | 0.279 | 0.219 | 0.263 | 0.258 | 0.288 | 0.244 | 0.286 | 0.233 | 0.273 | 0.233 | 0.273 | 0.238 | 0.281 | 0.224 | 0.264 |
| Solar | 96 | **0.160** | **0.193** | 0.181 | 0.242 | 0.179 | 0.248 | 0.192 | 0.239 | 0.205 | 0.241 | 0.196 | 0.258 | 0.217 | 0.255 | 0.232 | 0.271 | 0.205 | 0.239 |
| | 192 | **0.178** | **0.211** | 0.203 | 0.261 | 0.213 | 0.252 | 0.197 | 0.259 | 0.215 | 0.265 | 0.224 | 0.280 | 0.208 | 0.257 | 0.238 | 0.293 | 0.227 | 0.280 |
| | 336 | **0.190** | **0.218** | 0.219 | 0.272 | 0.222 | 0.261 | 0.212 | 0.273 | 0.213 | 0.276 | 0.240 | 0.288 | 0.238 | 0.309 | 0.234 | 0.301 | 0.225 | 0.290 |
| | 720 | **0.196** | **0.221** | 0.231 | 0.281 | 0.235 | 0.264 | 0.217 | 0.274 | 0.232 | 0.272 | 0.246 | 0.299 | 0.270 | 0.319 | 0.273 | 0.319 | 0.249 | 0.291 |
| | Avg | **0.181** | **0.211** | 0.209 | 0.264 | 0.216 | 0.254 | 0.201 | 0.264 | 0.216 | 0.264 | 0.227 | 0.281 | 0.233 | 0.285 | 0.244 | 0.296 | 0.227 | 0.275 |
| ETTh1 | 96 | 0.356 | 0.388 | 0.379 | 0.404 | **0.349** | **0.384** | 0.389 | 0.417 | 0.362 | 0.389 | 0.380 | 0.405 | 0.389 | 0.421 | 0.410 | 0.441 | 0.377 | 0.408 |
| | 192 | 0.397 | 0.416 | 0.429 | 0.441 | **0.387** | **0.410** | 0.427 | 0.443 | 0.404 | 0.412 | 0.418 | 0.428 | 0.424 | 0.446 | 0.448 | 0.465 | 0.413 | 0.431 |
| | 336 | 0.420 | 0.432 | 0.454 | 0.455 | **0.408** | **0.418** | 0.446 | 0.458 | 0.435 | 0.428 | 0.453 | 0.450 | 0.456 | 0.469 | 0.482 | 0.490 | 0.436 | 0.446 |
| | 720 | 0.432 | 0.456 | 0.499 | 0.506 | 0.439 | **0.446** | 0.468 | 0.491 | **0.426** | 0.448 | 0.480 | 0.484 | 0.545 | 0.532 | 0.475 | 0.500 | 0.455 | 0.475 |
| | Avg | 0.401 | 0.423 | 0.440 | 0.452 | **0.396** | **0.415** | 0.433 | 0.452 | 0.407 | 0.419 | 0.433 | 0.442 | 0.454 | 0.467 | 0.454 | 0.474 | 0.420 | 0.440 |
| ETTh2 | 96 | **0.269** | **0.329** | 0.288 | 0.354 | 0.292 | 0.345 | 0.309 | 0.365 | 0.340 | 0.346 | 0.273 | 0.341 | 0.305 | 0.361 | 0.315 | 0.380 | 0.276 | 0.339 |
| | 192 | **0.333** | **0.373** | 0.377 | 0.403 | 0.339 | 0.387 | 0.378 | 0.405 | 0.340 | 0.377 | 0.337 | 0.385 | 0.405 | 0.421 | 0.383 | 0.415 | 0.342 | 0.385 |
| | 336 | **0.357** | **0.396** | 0.377 | 0.415 | 0.358 | 0.410 | 0.460 | 0.461 | 0.360 | 0.398 | 0.369 | 0.414 | 0.411 | 0.436 | 0.385 | 0.438 | 0.364 | 0.405 |
| | 720 | **0.390** | **0.429** | 0.424 | 0.452 | 0.400 | 0.448 | 0.441 | 0.467 | 0.353 | 0.380 | 0.408 | 0.408 | 0.448 | 0.470 | 0.432 | 0.471 | 0.395 | 0.434 |
| | Avg | **0.337** | **0.382** | 0.367 | 0.406 | 0.347 | 0.398 | 0.397 | 0.425 | 0.344 | 0.387 | 0.347 | 0.397 | 0.392 | 0.422 | 0.379 | 0.426 | 0.344 | 0.391 |
| ETTm1 | 96 | **0.279** | **0.328** | 0.296 | 0.349 | 0.311 | 0.351 | 0.303 | 0.361 | 0.314 | 0.359 | 0.313 | 0.357 | 0.315 | 0.369 | 0.332 | 0.384 | 0.298 | 0.352 |
| | 192 | **0.323** | **0.358** | 0.337 | 0.374 | 0.338 | 0.371 | 0.336 | 0.377 | 0.348 | 0.376 | 0.343 | 0.377 | 0.349 | 0.388 | 0.355 | 0.398 | 0.335 | 0.373 |
| | 336 | **0.361** | **0.383** | 0.369 | 0.393 | 0.364 | 0.386 | 0.368 | 0.407 | 0.368 | 0.386 | 0.372 | 0.393 | 0.381 | 0.409 | 0.386 | 0.416 | 0.366 | 0.394 |
| | 720 | 0.421 | 0.416 | 0.447 | 0.434 | **0.413** | 0.414 | 0.438 | 0.438 | 0.419 | **0.413** | 0.420 | 0.420 | 0.437 | 0.439 | 0.452 | 0.457 | 0.420 | 0.421 |
| | Avg | **0.346** | **0.371** | 0.362 | 0.388 | 0.357 | 0.381 | 0.365 | 0.396 | 0.362 | 0.384 | 0.362 | 0.387 | 0.371 | 0.401 | 0.381 | 0.414 | 0.355 | 0.385 |
| ETTm2 | 96 | **0.155** | **0.240** | 0.169 | 0.257 | 0.167 | 0.259 | 0.188 | 0.274 | 0.167 | 0.259 | 0.179 | 0.269 | 0.179 | 0.274 | 0.192 | 0.285 | 0.165 | 0.260 |
| | 192 | **0.213** | **0.282** | 0.231 | 0.299 | 0.219 | 0.297 | 0.256 | 0.317 | 0.219 | 0.297 | 0.243 | 0.312 | 0.239 | 0.314 | 0.253 | 0.329 | 0.219 | 0.298 |
| | 336 | **0.267** | **0.319** | 0.282 | 0.337 | 0.271 | 0.330 | 0.334 | 0.366 | 0.271 | 0.330 | 0.270 | 0.330 | 0.309 | 0.356 | 0.307 | 0.362 | 0.268 | 0.333 |
| | 720 | **0.347** | **0.373** | 0.371 | 0.398 | 0.353 | 0.380 | 0.392 | 0.406 | 0.353 | 0.380 | 0.362 | 0.393 | 0.387 | 0.407 | 0.380 | 0.412 | 0.352 | 0.386 |
| | Avg | **0.246** | **0.304** | 0.263 | 0.323 | 0.253 | 0.317 | 0.293 | 0.341 | 0.253 | 0.317 | 0.264 | 0.326 | 0.279 | 0.338 | 0.283 | 0.347 | 0.251 | 0.319 |
| 1$^{st}$ Count | | 32 | 33 | 0 | 0 | 7 | 4 | 1 | 0 | 2 | 3 | 0 | 0 | 0 | 0 | 0 | 0 | 0 | 0 |

## 4.2 MAIN RESULTS

Table 2 summarizes the quantitative results for long-term time series forecasting across multiple prediction horizons and datasets. As shown, our proposed PMDformer achieves the lowest Mean Squared Error (MSE) and Mean Absolute Error (MAE) on 7 out of 8 real-world datasets, outperforming all baselines in the majority of cases. This success is directly tied to PMDformer's ability to overcome fundamental limitations in existing architectures.

Table 3: Ablation study on PMD module. We assess different modules for patch-wise normalization, along with removing PMD module. Results are averaged across all prediction horizons.

| Design | Norm | ETTh2 | | ETTm1 | | Weather | | Traffic | | Solar | |
|--------|------|-------|-----|-------|-----|---------|-----|---------|-----|-------|-----|
| | | MSE | MAE | MSE | MAE | MSE | MAE | MSE | MAE | MSE | MAE |
| PMDformer | PMD | **0.337** | **0.382** | **0.346** | **0.371** | **0.217** | **0.251** | **0.378** | **0.234** | **0.181** | **0.211** |
| Replace | w/ stdev | 0.354 | 0.392 | 0.347 | 0.370 | 0.218 | 0.252 | 0.396 | 0.259 | 0.205 | 0.221 |
| | SAN | 0.360 | 0.403 | 0.353 | 0.380 | 0.225 | 0.275 | 0.392 | 0.273 | 0.182 | 0.235 |
| | ✗ | 0.359 | 0.394 | 0.347 | 0.370 | 0.223 | 0.260 | 0.397 | 0.258 | 0.199 | 0.212 |

Table 4: Ablation studies on TRA and PVA modules in PMDformer: Performance impacts of replacements, removals, and order swaps across ETTh2, ETTm1, Traffic, and Solar datasets. Results are averaged across all prediction horizons.

| Design | TRA | PVA | ETTh2 | | ETTm1 | | Traffic | | Solar | |
|--------|-----|-----|-------|-----|-------|-----|---------|-----|-------|-----|
| | | | MSE | MAE | MSE | MAE | MSE | MAE | MSE | MAE |
| PMDformer | ✓ | Last Token | **0.337** | **0.382** | **0.346** | **0.371** | **0.378** | **0.234** | **0.181** | **0.211** |
| Replace | ✓ | All Token | 0.340 | 0.384 | 0.354 | 0.375 | 0.380 | 0.239 | 0.186 | 0.214 |
| | Self-attention | Last Token | 0.345 | 0.386 | 0.352 | 0.372 | 0.388 | 0.251 | 0.196 | 0.217 |
| | Swap Order ⇋ | | 0.342 | 0.385 | 0.350 | 0.372 | 0.379 | 0.235 | 0.188 | 0.216 |
| w/o | ✗ | Last Token | 0.344 | 0.381 | 0.347 | 0.372 | 0.410 | 0.270 | 0.215 | 0.226 |
| | ✓ | ✗ | 0.340 | 0.383 | 0.347 | 0.371 | 0.386 | 0.240 | 0.194 | 0.214 |
| | ✗ | ✗ | 0.346 | 0.384 | 0.351 | 0.372 | 0.426 | 0.288 | 0.222 | 0.230 |

Specifically, compared to the patch-based model TimeBase, PMDformer yields an average MSE reduction of 5.68% and MAE reduction of 6.61%. This improvement stems from our method's capacity to indentify meaningful shape similarities across patches, a capability that TimeBase's orthogonal patch selection inherently sacrifices to reduce redundancy. Moreover, against TQNet, PMDformer achieves an average MSE reduction of 8.62% and MAE reduction of 9.96%. TQNet's fixed periodic queries constrain its ability to handle diverse cycles, whereas PMDformer's adaptive proximal variable attention offers greater flexibility in modeling variables' shape similarities. Compared to the Transformer-based iTransformer, PMDformer delivers an average MSE reduction of 11.44% and MAE reduction of 12.38%. iTransformer captures dependencies among variable tokens embedded from the entire historical sequence, which can lead to overfitting on early, weakly relevant variable relationships that degrade future predictions. In contrast, our PVA module succeed to avoid this by focusing on the shape similarities of variables within the most nearest patch.

**PMD Module Analysis.** We assessed the effectiveness of the PMD module through extensive ablations conducted on five non-stationary benchmarks: ETTh2, ETTm1, Weather, Traffic, and Solar (Wen et al., 2023; Kim et al., 2025). Using a fixed input length of 720, we tested the model's performance across various prediction horizons (96, 192, 336, and 720) against several patch-wise normalization variants: (i) mean–variance standardization ('w/ stdev'), (ii) utilizing the Scale-Adaptive Normalization (SAN) (Liu et al., 2023b) method, and (iii) removing the PMD module entirely. As presented in Table 3, the PMDformer consistently achieves superior accuracy across all datasets. We attribute this advantage to the PMD module's per-patch centering mechanism, which effectively preserves crucial intra-patch shape information. This preservation allows the Transformer architecture to specifically concentrate its attention on modeling **shape similarity**. Furthermore, by explicitly injecting the patch mean as a separated trend component into the Transformer pathway, PMDformer is uniquely positioned to accurately capture and model long-term trends. In stark contrast, SAN explicitly decouples the scale and residual components for independent prediction. Since global scale estimation is inherently unstable in highly non-stationary series, this rigid decoupling undermines the essential joint modeling of scale–shape interactions, consequently leading to overfitting and weaker generalization capabilities.

**TRA & PVA Analysis.** To assess the effectiveness of the TRA and PVA modules, we conducted ablation studies on the ETTh2, ETTm1, Traffic, and Solar datasets. For the TRA module, we tested

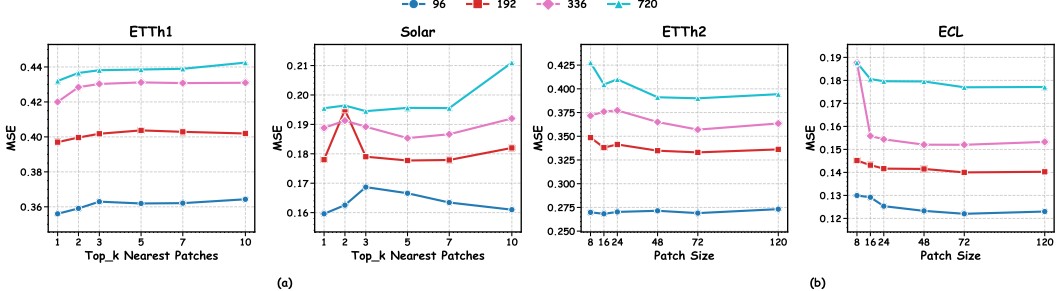

Figure 4: Parameter Sensitivity Analysis. (a) Selection of the number of $k$ nearest patches to the prediction sequence for capturing inter-variable dependencies on these patches. Superior and more stable performance is achieved when $k = 1$. (b) Different patch sizes are used to partition the input sequence, and a moderate patch size yields the optimal choice.

two alternatives: replacing it with standard self-attention or removing it entirely. For the PVA module, we either modified it to compute variable-wise shape similarity across *all* patches or removed the module completely. Additionally, we investigated a structural variant that swaps the sequential order of the two modules. The experimental outcomes are summarized in Table 4.

The results unequivocally show that PMDformer consistently outperforms all ablated variants across every dataset and configuration. When TRA is replaced with standard self-attention, performance degrades significantly because the crucial long-term trend information is neglected. Similarly, when PVA is forced to compute variable-wise shape similarity across all historical patches, performance decreases. This confirms our hypothesis that early variable relationships are often only weakly or spuriously correlated with the predictive sequences, justifying PVA's proximal focus. Furthermore, removing both TRA and PVA results in the largest performance drop observed, emphatically highlighting the dual importance of TRA in modeling temporal patch shapes and long-range trends, and PVA in capturing relevant variable-wise shape similarity. Finally, swapping the original order of TRA and PVA also causes notable performance degradation. When TRA is applied first, it compresses patch information too early, making it harder for the subsequent variable modeling to identify meaningful cross-variable dependencies.

## 4.3 PARAMETER SENSITIVITY ANALYSIS

**Patch Count for Cross-Variable Modeling.** We evaluate the impact of capturing variable patterns within different numbers of patches, where $k \in \{1, 2, 3, 5, 7, 10\}$. For each setting, the $k$ nearest patches to future sequences are selected to capture the shape similarity of variable, thereby further validating the effectiveness of PVA. Experiments are conducted on the ETTh1 and Solar datasets. As shown in Figure 4 (a), the mean squared error (MSE) exhibits an overall upward trend as $k$ gradually increases on the ETTh1 dataset. On the Solar dataset, this increase is more pronounced when predicting 192, 336, or 720 steps ahead, because future sequences are more weakly correlated with early variable relationships. Moreover, the MSE curves show some fluctuations, indicating that different values of $k$ may lead to more significant differences in prediction performance. In contrast, across all four prediction horizons, using $k = 1$ yields more stable performance compared with larger $k$. This is because the nearest patch is typically more closely aligned with the target sequence to be predicted, making it more beneficial for accurate modeling.

**Patch Size.** Different patch sizes lead to varying degrees of distinction among patches. To investigate this, we evaluate multiple patch sizes $\{8, 16, 24, 48, 72, 120\}$ on the ETTh2 and ECL datasets. As shown in Figure 4 (b), both overly small and overly large patch sizes fail to deliver optimal performance. This is because excessively small patches provide insufficient shape information to distinguish similarity, making it difficult for the attention mechanism to capture underlying temporal dependencies or genuine variable correlations. Conversely, overly large patches reduce the number of tokens, thereby limiting the model's ability to capture long-range dependencies. Based on these observations, we find that moderate patch sizes, particularly within $\{24, 48, 72\}$, achieve a better trade-off and yield more robust performance.

## 5 CONCLUSION

In this paper, we tackle challenges in long-term time series forecasting by emphasizing true shape similarities hidden by scale variations in non-stationary data. Our patch-mean decoupling (PMD) separates trends from residual shapes while preserving amplitudes, enabling shape-focused attention across patches and variables. Integrated with proximal variable attention (PVA) for recent inter-variable dependencies and trend restoration attention (TRA) for global trend reintegration. Experiments on LTSF benchmarks show PMDformer surpasses state-of-the-art baselines in accuracy and stability, underscoring the value of shape-centric Transformer designs. Future directions include scaling to higher-dimensional multivariate data and multimodal integrations for applications in energy, finance, and traffic.

## 6 ACKNOWLEDGEMENTS

This work was supported by the Major Science and Technology Special Project of the Sichuan Provincial Department of Science and Technology (Grant No. 2024ZDZX0002), the Sichuan Provincial Innovation Group Project (Grant No. 2024NSFTD0054), Fundamental Research Funds for the Central Universities (JBK202511081), the Blockchain Research Center of China, the Natural Science Foundation of China (Grant No. 62502397), the National Natural Science Foundation of China (Grant No. 72471197), and the Sichuan Provincial Philosophy and Social Science Fund (Grant No. SCJJ25ND091).

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

## A APPENDIX

### A.1 EFFICIENCY ANALYSIS

To evaluate the efficiency of our model in handling complex tasks, we conduct experiments under two settings: varying the number of variables and varying the input length. In the first setting, we fix the input length at 720 and change the number of variables; in the second setting, we fix the number of variables at 100 and test PMDformer with different input lengths. The batch size is set to 1 in all experiments. The results are shown in Figure 5. Under both settings, compared with recent popular models such as PatchTST (Nie et al., 2023), iTransformer (Liu et al., 2024a), and ModernTCN (Luo & Wang, 2024), PMDformer requires significantly less GPU memory, thereby reducing the overall computational cost.

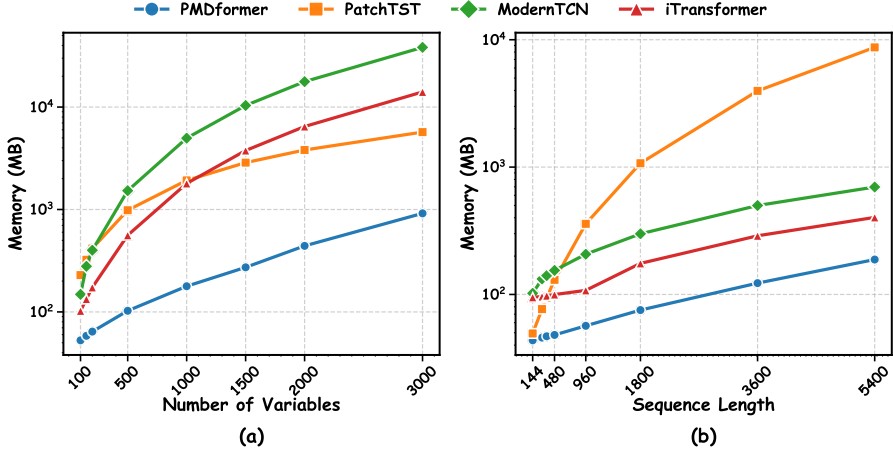

Figure 5: (a) Comparison of memory usage with varying number of variables $C$. (b) Comparison of memory usage with varying input sequence length $L$. PMDformer consistently requires the lowest memory.

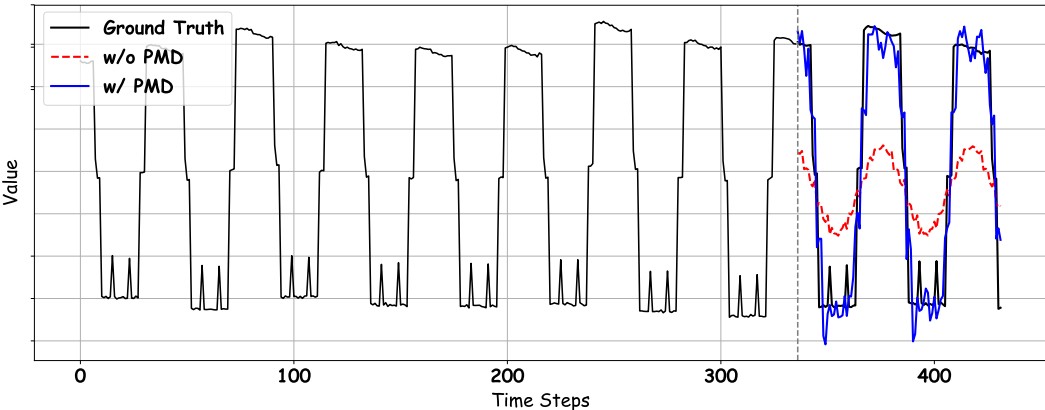

Figure 6: Comparison on synthetic data. The ground truth alternates between pulse and sine shapes with varying scales. The 'w/o PMD' yields smoothed outputs and struggles to recognize the shape similarity, while 'w/ PMD' effectively fits the shapes and trends.

### A.2 COMPARISON ON SYNTHETIC DATA

To further validate the effectiveness of our PMD module, we conduct an experiment on a synthetic dataset. This dataset consists of patches alternating between two different shapes: a sharp pulse wave

with large amplitude and a smooth sine wave with small amplitude. To simulate non-stationary time series, the patches exhibit varying scales and are augmented with moderate noise. We compare a standard patch-based Transformer (w/o PMD) against our model incorporating the patch-mean decoupling module (w/ PMD). As illustrated in Figure 6, the 'w/o PMD' model struggles to recognize true shape similarities due to scale differences between patches, leading to predictions that resemble mostly smooth curves with inadequate trend fitting. In contrast, our 'w/ PMD' model, by removing scale factors, enables attention to focus more effectively on intrinsic shapes, resulting in predictions that better capture both the underlying patterns and long-range trends.

## B    ON THE USE OF LARGE LANGUAGE MODELS

The authors used large language models (LLMs) exclusively for language polishing and minor rephrasing during the final writing stage. All scientific content, ideas, and initial drafts were created entirely by the authors without any text improved by LLMs was carefully checked and edited by the authors. LLMs played no role in developing research questions, designing experiments, analyzing results, or any other aspect of the research itself.

