# OpenReview forum: "PMDformer: Patch-Mean Decoupling Information Transformer for Long-term Forecasting"
_ICLR.cc/2026/Conference — ICLR 2026 Poster_

### Official Review · Reviewer_1ngW · 2025-10-28

**Soundness:** 3
**Presentation:** 3
**Contribution:** 3
**Rating:** 6
**Confidence:** 5

**Summary:**

This paper proposes PMDFormer, a novel Transformer-based model for Long-term Time Series Forecasting (LTSF). The key insight is that the attention mechanism in existing patch-based models is often biased by the scale (amplitude) of patches, hindering its ability to capture true shape similarities. To address this, the authors introduce three core components: 1) Patch-Mean Decoupling (PMD), which centers patches by subtracting their mean to separate trend from shape; 2) Proximal Variable Attention (PVA), which focuses cross-variable modeling only on the most recent patch to avoid noisy historical correlations; and 3) Trend Restoration Attention (TRA), which reintegrates the global trend back into the attention mechanism. Extensive experiments on eight benchmarks show that PMDFormer achieves state-of-the-art performance, outperforming recent strong baselines. Ablation studies and theoretical analysis validate the design of each component.

**Strengths:**

1. Novel and well-motivated core idea. The identification of "scale bias" in patch-based attention is insightful.

2. The model demonstrates compelling state-of-the-art performance, outperforming a wide range of recent and strong baselines across eight standard benchmarks.

3. The paper provides thorough ablation studies and a theoretical analysis that convincingly justify the contribution of each proposed module and the underlying motivation.

**Weaknesses:**

1. The paper positions PMDFormer as effectively modeling cross-variable dependencies, a domain where many previous "Variable-Dependent" (VD) models have struggled. However, the PVA module is applied only to the most recent patch. While the results are excellent, this design choice essentially limits cross-variable modeling to a very short, recent context. The paper could more explicitly discuss the implications of this: is the success of PMDFormer evidence that long-range cross-variable dependencies are generally not useful for LTSF, or that they are too noisy to model effectively? A comparison of PVA's performance when applied to the last $K$ patches (beyond just the ablation on $k$ in the sensitivity analysis) could have deepened this analysis.

2. While the efficiency of PVA is mentioned, a more formal and overall complexity analysis of the full PMDFormer architecture compared to other leading models (e.g., PatchTST, iTransformer, TimeBase) is missing. Given the use of a Transformer encoder in the TRA module and the separate PVA module, a discussion of the total parameter count and FLOPs would be beneficial for a complete picture.

3. The figure references "Figure X" in the Patch Size sensitivity analysis (Page 9), which appears to be a placeholder. This should be corrected to the appropriate figure number (likely 4b).

**Questions:**

1. The PVA module operates on the embedded tokens after PMD. Given that PMD centers the patches, does this mean PVA is exclusively modeling the shape similarities of variables at the most recent time segment, completely independent of their absolute levels? Could there be scenarios where the absolute values (or scales) of variables in the proximal patch are also critical for prediction?

2. The TRA module reintegrates the trend via a simple broadcast addition to the Value projection (Eq. 8). Was there an exploration of more complex fusion mechanisms (e.g., gating, concatenation followed by a linear layer)? Why was additive reintegration chosen as the most effective method?

3. For large multivariate datasets (e.g., Traffic with 862 variables), can the authors provide detailed training/inference time and memory usage comparisons between PMDformer and baselines (e.g., iTransformer, TimeBase)? How does PMDformer’s efficiency scale with the number of variables (C) or input sequence length (L), and is there potential to further optimize the TRA module’s computation (e.g., via parameter sharing)?

4. The paper finds moderate patch sizes (24–72) are optimal, but what guidelines would the authors recommend for selecting patch size based on dataset properties (e.g., sampling interval, length of input sequence, seasonality period)? For example, should a dataset with 10-minute sampling intervals (e.g., Weather) use a smaller patch size than one with 1-hour intervals (e.g., ECL)?

---

> ### Author Response · Authors · 2025-11-21
> **Responses to Reviewer 1ngW (Part 1/4)**
>
> # W1:
> **Response:**  We thank the reviewer for this insightful question regarding the temporal scope of cross-variable modeling.
>
> ​	(1). We claim that long-range cross-variable dependencies are generally not useful for LTSF. Cross-variable relationships are often **non-stationary** and **evolve over time**, so the recent interactions are typically the most predictive of future behavior. For instance, in stock markets, asset correlations often remain low in calm periods but spike sharply during crise. This phenomena indicates that recent data could better capture the prevailing trend. On the other hand, incorporating long-range cross-variable dependencies across the entire historical context introduces **substantial redundancy and noise**, which can degrade performance. While a small subset of earlier interactions may occasionally remain informative, capturing these effectively would require **adaptive selection** mechanisms (e.g., inspired by the information bottleneck principle), which we leave to future work.
>
> ​	(2). We conducted an additional ablation applying PVA to the last K patches (K=1,2,3,4). Performance peaks at K=1, confirming that extending cross-variable modeling beyond the most recent patch does **not yield benefits**. We will add these results to the revised manuscript and updated the discussion accordingly.
>
> |       | last_1 MSE | last_2 MSE | last_3 MSE | last_4 MSE |
> | :---: | :--------: | :--------: | :--------: | :--------: |
> | ETTh1 | **0.401**  |   0.403    |   0.408    |   0.408    |
> | Solar | **0.181**  |   0.186    |   0.183    |   0.183    |

---

> ### Author Response · Authors · 2025-11-21
> **More Responses to Reviewer 1ngW (Part 2/4)**
>
> # W2:
> **Response:**  Thank you for this constructive feedback. We have conducted a comprehensive efficiency evaluation on multiple datasets including ETTm1, Weather, ECL, and Traffic.
>
> ​	**metrics** : FLOPs, Parameter count, Memory Usage, Inference Iime and Training Time.
>
> ​	**Baselines**: This evaluation compares PMDFormer against leading baselines including PatchTST, iTransformer, TimeBase, and ModernTCN. Notably, TimeBase is a non-Transformer lightweight model.
>
> ​	 **Experimental setting**:  All models were evaluated with a fixed input length of 720 on an NVIDIA A100 (40 GB). Inference time uses batch size 1 and is averaged over 5 runs; training time is averaged over 5 epochs. Results are averaged across the four standard prediction horizons, with MSE included only as a performance reference.
>
> ​	PMDFormer consistently uses **far less memory** and achieves substantially **faster** training and inference than PatchTST and ModernTCN while delivering better prediction accuracy. Relative to iTransformer, PMDFormer requires fewer parameters across almost all datasets and achieves superior prediction performance, though it incurs higher FLOPs on highly multivariate datasets (ECL, Traffic) that necessitate **additional computation for capturing complex temporal interactions and achieving enhanced accuracy**. TimeBase, being a non-Transformer lightweight model, exhibits the lowest parameter count and fastest inference on most datasets but longer inference on Weather due to its independent prediction heads per channel. And its extreme minimalism, although highly efficient, generally limits the peak performance. Overall, PMDFormer offers the most favourable balance between efficiency and accuracy among Transformer-based baselines.
>
> |        ETTm1        | PMDformer | PatchTST | ModernTCN | iTransformer | TimeBase |
> | :-----------------: | :-------: | :------: | :-------: | :----------: | :------: |
> |        FLOPS        |  27.88M   |  2.22G   |   2.55G   |    29.42M    |  1.16M   |
> |       Params        |   0.18M   |  4.27M   |   5.50M   |    0.34M     |  4.67K   |
> |  Memory Usage: MB   |   23.22   |  340.24  |  452.84   |    23.94     |  18.19   |
> | Inference Time: ms  |   2.08    |   2.24   |   3.14    |     1.64     |   0.47   |
> | Train Time: s/epoch |   74.16   |  72.80   |  105.92   |    44.23     |  27.03   |
> |         MSE         |   0.346   |  0.355   |   0.362   |    0.371     |  0.357   |
>
> |       Weather       | PMDformer | PatchTST | ModernTCN | iTransformer | TimeBase |
> | :-----------------: | :-------: | :------: | :-------: | :----------: | :------: |
> |        FLOPS        |   0.20G   |  6.67G   |   3.44G   |    1.05G     |  1.06M   |
> |       Params        |   0.29M   |  4.27M   |   5.80M   |    5.28M     |  0.04M   |
> |  Memory Usage: MB   |   43.17   |  855.87  |  506.67   |    119.35    |  20.52   |
> | Inference Time: ms  |   2.15    |   2.29   |   1.81    |     2.24     |   3.38   |
> | Train Time: s/epoch |   78.15   |  85.53   |   77.07   |    71.30     |  135.43  |
> |         MSE         |   0.217   |  0.224   |   0.233   |    0.233     |  0.219   |
>
> |         ECL         | PMDformer | PatchTST | **ModernTCN** | **iTransformer** | **TimeBase** |
> | :-----------------: | :-------: | :------: | :-----------: | :--------------: | :----------: |
> |        FLOPS        |  32.45G   | 102.00G  |    194.54G    |      13.71G      |    16.27M    |
> |       Params        |   2.99M   |  4.27M   |    132.03M    |      5.28M       |    0.28K     |
> |  Memory Usage: MB   |  1107.74  | 11990.55 |    7938.11    |      481.55      |    59.69     |
> | Inference Time: ms  |   2.79    |   9.59   |     16.93     |       2.58       |     1.49     |
> | Train Time: s/epoch |   67.36   |  440.04  |    262.70     |      60.18       |    32.05     |
> |         MSE         |   0.148   |  0.169   |     0.158     |      0.166       |    0.167     |
>
> |       Traffic       | PMDformer | PatchTST | ModernTCN | iTransformer | TimeBase |
> | :-----------------: | :-------: | :------: | :-------: | :----------: | :------: |
> |        FLOPS        |  294.34G  | 273.92G  | 1210.06G  |    47.46G    |  68.19M  |
> |       Params        |   6.89M   |  4.27M   |  825.54M  |    6.85M     |  0.44K   |
> |  Memory Usage: MB   |  5429.63  | 32090.30 | 27677.18  |   2658.48    |  128.06  |
> | Inference Time: ms  |   4.41    |  21.41   |   45.94   |     3.56     |   1.32   |
> | Train Time: s/epoch |  243.60   |  836.69  |  725.83   |    80.24     |  41.42   |
> |         MSE         |   0.378   |  0.394   |   0.408   |    0.407     |  0.418   |
>
> # W3:
> **Response**：We thank the reviewer for identifying this oversight. We have revised the manuscript to replace the placeholder "Figure X" with "Figure 4 (b)" in the Patch Size sensitivity analysis on page 9.

---

> ### Author Response · Authors · 2025-11-21
> **More Responses to Reviewer 1ngW (Part 3/4)**
>
> # Q1:
> **Response**:
>
> ​	(1). The reviewer is **correct** that PVA, by operating on PMD-centered patches, exclusively models shape similarities while **disregarding** the absolute values of variables in the proximal patch. This design choice intentionally **prevents scale differences** from distorting the estimation of true cross-variable correlations.
>
> ​	(2). In multivariate time series forecasting, the useful cross-variable relationships for accurate predictions are typically driven by **shared temporal patterns (shapes)** rather than absolute magnitudes. Our choice to ignore absolute levels is therefore deliberate and aligns with the nature of most benchmarks.
>
> ​	We conducted an ablation study on the ETTh2, Traffic, and Solar-Energy. We modified PVA to incorporate the absolute values in the proximal patch while keeping all other components unchanged. The results averaged across the all horizons are as follows:
>
> |         | PMDformer |           | + Var_ab |       |
> | :-----: | :-------: | :-------: | :------: | :---: |
> |         |    MSE    |    MAE    |   MSE    |  MAE  |
> |  ETTh1  | **0.401** | **0.423** |  0.406   | 0.426 |
> | Weather | **0.217** | **0.251** |  0.222   | 0.260 |
> | Traffic | **0.378** | **0.234** |  0.384   | 0.240 |
>
> ​	Incorporating absolute values consistently degrades performance, confirming that modeling shapes alone is more beneficial for these datasets.  And we have not yet identified a scenario in which the absolute values of variables in the proximal patch are critical for prediction.
>
> # Q2:
> **Response:**
>
> ​	We thank the reviewer for this insightful question.
>
> ​	(1). We conducted extensive ablation studies comparing the additive fusion in Equation 8 against alternatives including **linear projection**, **gating mechanisms**, and **concatenation followed by a linear layer**. These evaluations spanned multiple datasets and forecasting horizons, with averaged results presented in the table below. The additive approach consistently outperformed or matched the alternatives in prediction accuracy, demonstrating its robustness.
>
> |         | PMDformer |           | Linear_Projection |       | Gating_Projection |       | Concat_Linear |           |
> | :-----: | :-------: | :-------: | :---------------: | :---: | :---------------: | :---: | :-----------: | :-------: |
> |         |    MSE    |    MAE    |        MSE        |  MAE  |        MSE        |  MAE  |      MSE      |    MAE    |
> |  ETTh2  | **0.337** | **0.382** |       0.341       | 0.383 |       0.343       | 0.384 |     0.344     |   0.386   |
> |  ETTm1  | **0.346** | **0.371** |       0.369       | 0.385 |       0.354       | 0.374 |     0.350     |   0.373   |
> | Traffic | **0.378** | **0.234** |       0.381       | 0.239 |       0.386       | 0.249 |     0.392     |   0.257   |
> |  Solar  | **0.181** |   0.211   |       0.186       | 0.211 |       0.184       | 0.213 |     0.182     | **0.210** |
>
> ​	(2). The additive reintegration was ultimately chosen because it is directly inspired by **residual connections**: it reintroduces the trend with minimal computational overhead and no additional parameters. The ablation studies above confirm that this simple yet effective design matches or outperforms more complex fusion mechanisms in practice.
>
> # Q3:
> **Response:** Thank you for the valuable suggestion on efficiency analysis.
>
> ​	(1). The comparison on Traffic please refer to Weakness 2.
>
> ​	(2). Results with respect to both the number of variables $C$ and input sequence length $L$ are provided in the revised appendix (**new Figure 5**). The figure demonstrates that PMDformer exhibits **lower memory usage** compared to leading models (PatchTST, iTransformer, ModernTCN).
>
> ​	(3).We agree that additional efficiency gains are possible through linear attention mechanisms or FlashAttention-2/3. We adopted the same standard self-attention implementation as PatchTST and iTransformer in the current experiments to ensure fair comparison. More advanced attention optimizations will be explored in future work.

---

> ### Author Response · Authors · 2025-11-21
> **More Responses to Reviewer 1ngW (Part 4/4)**
>
> # Q4:
> **Response:**
>
> ​	We thank the reviewer for this practical and insightful question. Based on recent studies [1, 2], we select a patch size that aligns with the dataset’s intrinsic **seasonality period** and **sampling interval**. A useful guideline is to choose a patch size that is either a **divisor** or a **multiple** of the fundamental periodicity (e.g., 24, 48 or 72 for hourly data with a daily cycle). This alignment ensures that each patch captures a semantically coherent temporal pattern.
>
> ​	In cases where the seasonality is unclear or difficult to estimate, practitioners may rely primarily on the sampling interval. As a rule of thumb, smaller patch sizes tend to work better for high-frequency datasets, as they allow the model to more effectively capture short-term fluctuations.
>
> [1] Wang Y, Qiu Y, Chen P, et al. LightGTS: A Lightweight General Time Series Forecasting Model[J]. arXiv preprint arXiv:2506.06005, 2025.
>
> [2] Chen M, Shen L, Li Z, et al. Visionts: Visual masked autoencoders are free-lunch zero-shot time series forecasters[J]. arXiv preprint arXiv:2408.17253, 2024.

---

> > ### Comment · Reviewer_1ngW · 2025-11-26
> >
> > Thanks to the author's detailed response, which has resolved my questions.
> > Therefore, I will keep my positive rating unchanged.

---

> > > ### Author Response · Authors · 2025-11-26
> > > **Response to Reviewer 1ngW**
> > >
> > > Thank you very much for your thoughtful feedback and for maintaining your positive rating. We appreciate your time and consideration!

---

### Official Review · Reviewer_rABD · 2025-10-28

**Soundness:** 3
**Presentation:** 3
**Contribution:** 3
**Rating:** 6
**Confidence:** 4

**Summary:**

This paper proposes PMDformer, targeting the core issue that shape matching is often dominated by scale. It adopts Patch-Mean Decoupling (PMD), which removes only the mean of each patch, applies Proximal Variable Attention (PVA) to perform cross-variable attention on the nearest patch to the forecasting window, and uses Trend Restoration Attention (TRA) to inject trend information back into the Value branch without disturbing the Q/K shape alignment, thereby achieving a unified design that separates and then fuses “shape” and “trend.” Across eight datasets and multiple forecasting horizons, PMDformer shows stable improvements over strong baselines, and ablation studies validate the necessity of the three modules and their ordering.

**Strengths:**

S1. The problem is clearly defined, and each module is well-motivated by the design goals.

S2. The paper includes a formal derivation of the “mean-dominance” condition and multi-module ablations; the experimental design is fairly complete, and PMDformer performs well.

S3. The presentation is clear and the code structure is easy to follow.

**Weaknesses:**

W1. PMD is conceptually close to RevIN (reversible instance normalization that removes mean/variance per series) and to decomposition-based approaches such as DLinear and Autoformer (alleviating scale/distribution shift or decoupling trend and seasonality). The paper could strengthen the analysis to emphasize the differences.

W2. Restricting cross-variable modeling to only the nearest patch helps suppress noise and overfitting, but may lose interpretable dependencies when long-lag cross-variable effects exist.

W3. The similarity of patch-level trends depends heavily on the patch segmentation scheme, yet the paper does not analyze this sensitivity.

**Questions:**

Q1. After injecting trend into the Value branch, could residual pathways indirectly affect the attention output and thus interfere with shape alignment?

Q2. Beyond a fixed k=1, can a gated/learned scheduler adaptively determine the size of the proximal window or whether to include cross-variable attention over earlier patches?

Q3. How do you rule out—or control for—the confounding effects introduced by the patch segmentation choice?

---

> ### Author Response · Authors · 2025-11-21
> **Responses to Reviewer rABD (Part 1/2)**
>
> # W1:
>
> **Response:** We thank the reviewer for this insightful comment. PMD is **a bias correction strategy** tailored to the patch-based attention mechanism, distinct from data normalization or decomposition techniques. The key differences are outlined below.
>
> |          Method           |                          PMD (Ours)                          |                            RevIN                             |                 DLinear / Autoformer                  |
> | :-----------------------: | :----------------------------------------------------------: | :----------------------------------------------------------: | :---------------------------------------------------: |
> |         **Scope**         |                          Per-patch                           |                          Per-series                          |                      Per-series                       |
> |         **Goal**          | Remove scale bias in $Q$/$K$ attention to focus on shape similarity (per-patch) | Improve generalization by addressing distribution shift (per-series) |   Separate slow-varying trend from other components   |
> |    **Core Operation**     |                     $P_j^i - {\mu}_j^i$                      |                   $\frac{X - \mu}{\sigma}$                   |          $X_{Seasonal} = X - \text{Pool}(X)$          |
> | **Relation to Attention** |             scale addition to the Value pathway              |    A pre/post-processing module independent of attention     | A pre/post-processing module independent of attention |
>
>
> # W2:
>
> **Response:**
>
> ​	We acknowledge that, in theory, restricting cross-variable attention to only the nearest patch in PVA could potentially overlook long-lag dependencies in domains where such effects are prominent. However, our design prioritizes recent interactions because time series relationships are often non-stationary, with proximal dependencies being most indicative of future behavior.
>
> ​	Models that perform full historical cross-variable modeling, such as iTransformer and ModernTCN, **indeed attempt to capture long-lag effects** but consistently achieve **lower empirical performance** than our approach across standard benchmarks. Capturing genuine long-lag dependencies while simultaneously suppressing noise from outdated or spurious correlations remains **extremely challenging**, and existing methods that pursue unrestricted cross-variable attention generally sacrifice overall forecasting accuracy for marginal gains in interpretability.
>
> ​	Our method therefore offers a more **practical** and higher-performing solution under current evaluation protocols. In future work, we plan to explore **adaptive mechanisms** that selectively attend to meaningful long-lag cross-variable relations when evidence supports their relevance.
>
> # W3:
> **Response:** Thank you for this insightful comment. Our patch size sensitivity analysis in **Figure 4 (b)** of the manuscript addresses this concern. The results show that extreme patch sizes degrade performance, whereas the model maintains robust performance over a broad range of moderate patch sizes from 24 to 72. Within this range, the performance curve remains **stable** without significant fluctuations, indicating that patch-level trend similarities and overall model performance are **not critically sensitive** to precise segmentation choices. Intuitively，we select a patch size that aligns with the dataset’s intrinsic **seasonality period** and **sampling interval**.
>
> # Q1:
> **Response:** Thank you for the insightful question.
>
> ​	In the TRA module, multi-head self-attention follows the standard computation order:
>
> 1. $Q$ and $K$ are computed solely from the shape-only input $P^i$, so attention weights $\mathcal{A} = \text{Softmax}\Big(\frac{\mathbf{Q}^i (\mathbf{K}^i)^\top}{\sqrt{d}}\Big)$ depend exclusively on shape similarity.
> 2. Trend is injected only into the Value branch, yielding ${\hat{V}^i}:=V^i+\mu^i$.
> 3. The attention output is $\mathcal{A}\hat{V}^i$, which carries the trend information.
> 4. The residual connection then adds this output to the original input: $H^i=P^i+ Linear(\mathcal{A}\hat{V}^i)$.
>
> ​	Since attention weights $\mathcal{A}$ are finalized before the trend-injected $\hat{V}$ is applied, and the residual addition occurs only afterward, there exists **no pathway** for the trend to influence the attention weights in the current layer. The trend affects solely the final token representations, leaving shape-driven attention completely undisturbed.
>
> ​	This design achieves exactly the intended behavior described in Section 3.2. In contrast, injecting trend into $Q$ or $K$ would directly bias attention toward magnitude similarity—which our value-branch injection explicitly avoids.

---

> ### Author Response · Authors · 2025-11-21
> **More Responses to Reviewer rABD (Part 2/2)**
>
> # Q2:
> 1. **Response:** Thank you for this excellent and insightful suggestion.
>
>    (1). Following your recommendation, we implemented an **adaptive gated mechanism** (Gate_Var) that learns whether to include cross-variable attention for earlier patches beyond the fixed proximal window (k=1). The gating network takes the features of all patches as input and produces a **probability** for each patch. If the probability exceeds a predetermined threshold, cross-variable attention is applied to that patch, enabling adaptive selection of distant patches. The averaged results across all prediction lengths are shown below.
>
>    |         | PMDformer |           | Gate_Var |       |
>    | :-----: | :-------: | :-------: | :------: | :---: |
>    |         |    MSE    |    MAE    |   MSE    |  MAE  |
>    |  ETTh2  | **0.337** | **0.382** |  0.340   | 0.382 |
>    |  ETTm1  | **0.346** | **0.371** |  0.349   | 0.373 |
>    | Traffic | **0.378** | **0.234** |  0.387   | 0.246 |
>    |  Solar  | **0.181** | 0.211 |  0.185   | **0.210** |
>
>
>
>    ​	Gate_Var does not improve over the original model. We attribute this to the **absence** of a dedicated loss that explicitly encourages the gate to **suppress redundant cross-variable interactions**. Your suggestion is highly valuable. Exploring a more effective adaptive scheduler with proper regularization or auxiliary losses is a promising direction for future work and could be a new work.
>
>    (2). We also empirically evaluated larger fixed proximal windows (last $K$ patches, $K \in \{1,2,3,4\}$). The results shown in the table below indicates that attending only to the most recent patch is optimal in our setting.
>
> |       | last_1 MSE | last_2 MSE | last_3 MSE | last_4 MSE |
> | :---: | :--------: | :--------: | :--------: | :--------: |
> | ETTh1 | **0.401**  |   0.403    |   0.408    |   0.408    |
> | Solar | **0.181**  |   0.186    |   0.183    |   0.183    |
>
> # Q3:
> **Response:** Thank you for this question.
>
> ​	Based on recent studies [1, 2], we select a patch size that aligns with the dataset’s intrinsic **seasonality period** and **sampling interval**. A useful guideline is to choose a patch size that is either a **divisor** or a **multiple** of the fundamental periodicity (e.g., 24, 48 or 72 for hourly data with a daily cycle). This alignment ensures that each patch captures a semantically coherent temporal pattern.
>
> ​	We conducted a sensitivity analysis across multiple patch sizes $\{8, 16, 24, 48, 72, 120\}$ on ETTh2 and ECL datasets  (Figure 4 (b) in manuscript). The results show that PMDFormer achieves robust performance over a range of moderate patch sizes (e.g., 24 to 72), rather than depending on a single optimal value. This robustness indicates that the performance gains are are attributable to our method's design rather than a specific patch segmentation choice.
>
> [1] Wang Y, Qiu Y, Chen P, et al. LightGTS: A Lightweight General Time Series Forecasting Model[J]. arXiv preprint arXiv:2506.06005, 2025.
>
> [2] Chen M, Shen L, Li Z, et al. Visionts: Visual masked autoencoders are free-lunch zero-shot time series forecasters[J]. arXiv preprint arXiv:2408.17253, 2024.

---

### Official Review · Reviewer_pzqK · 2025-10-29

**Soundness:** 2
**Presentation:** 3
**Contribution:** 3
**Rating:** 6
**Confidence:** 4

**Summary:**

This work proposes a Transformer model named PMDformer for Long-Term Time Series Forecasting. The core idea is to decouple the trend and shape information of each time patch via Patch-Mean Decoupling (PMD), thereby preserving the original amplitude structure and avoiding the distortion of shape information caused by traditional normalization methods. Furthermore, the model introduces the Proximal Variable Attention (PVA) and Trend Restoration Attention (TRA) modules, which are designed to capture the most relevant short-term dependencies among variables and restore global trend information, respectively.

**Strengths:**

1. The ideas of this work are easy to understand. The description and presentation are clear.
2. This work conducts ablation experiments for the three core modules—PMD, PVA, and TRA—on multiple datasets, validating the effectiveness of each module.

**Weaknesses:**

1. The foundational premise of the PMD module is not fully convincing. According to Figure 1, the original patches (P1, P2), which have more similar means, receive a lower attention score than (P1, P3). This observation appears to contradict the authors' claim that "the scale differences initially obscure true shape similarity", as the patches with similar scales (P1, P2) are not assigned higher attention. This inconsistency raises questions about the necessity and motivation of the proposed decoupling.
2. In the TRA module, the operation of adding μ to V is not sufficiently motivated. Directly adding the raw, unprojected trend mean μ to the projected value V forcibly mixes vectors from disparate spaces—the original data space and the projected feature space.
3. The "double addition" of the trend in the architecture appears redundant. The trend term μ is added in the TRA module and again at the final projection layer. This design is not well-principled and risks over-emphasizing the trend component, potentially distorting the learned representations.

**Questions:**

See Weaknesses.

---

> ### Author Response · Authors · 2025-11-21
> **Responses to Reviewer pzqK**
>
> # W1:
>
> **Response:** Thank you for the careful reading and the insightful comment.
>
> ​	As illustrated in Figure 1, $P_1$,  $P_2$ and $P_3$ actually have distinctly **different means** rather than similar means, as indicated by their different colored dashed lines (blue for $P_1$, maroon for $P_2$, and orange for $P_3$). The core motivation in fact is convincing, our original explanation may have caused some misunderstanding, and we have revised the description to enhance clarity.
>
> ​	The attention weights are influenced by both shape and scale, making it **difficult** for them to reflect the true shape similarity—which is our primary interest in time series data. As shown in Figure 1, **$P_1$,  $P_2$ and $P_3$ have different scales (means)**. $P_1$ and $P_2$ are more similar in shape，while $P_3$ is structurally dissimilar. However, the attention score between $P_1$ and $P_3$ is higher than that between $P_1$ and $P_2$. This indicates that **scale differences initially obscure true shape similarity**. After applying PMD to remove the scale of patches, the attention score between $P_1$ and $P_3$ is lower than that between $P_1$ and $P_2$,  which aligns with their actual shape similarity.
>
>
>
> # W2:
> **Response:** Thank you for this insightful comment.
>
> ​	The additive reintegration of $\mu$ into $V$ is directly inspired by residual connections in ResNet. It reintroduces the trend with minimal computational overhead and no additional parameters. We conducted additional ablation studies comparing our method with alternatives.
>
> ​	Experimental setting: (i) **Linear_Projection**: linear projection of $\mu$ before addition, (ii) **Gating_Projection**: gating mechanisms, and (iii) **Concat_Linear**: concatenation of the projected Value and $\mu$ followed by a linear layer.
>
> ​	The averaged results across all prediction lengths are shown in the table below. PMDformer consistently matches or outperforms the alternatives. This **empirically validates** that injecting the raw trend μ without any projection or non-linear transformation is highly effective and avoids potential distortion of the globally extracted trend, which is crucial for long-term forecasting accuracy. We will clarify this motivation in the revised manuscript.
>
> |      Method       | ETTh2 MSE | ETTm1 MSE | Traffic MSE | Solar MSE |       Params       |
> | :---------------: | :-------: | :-------: | :---------: | :-------: | :----------------: |
> |     PMDformer     | **0.337** | **0.346** |  **0.378**  | **0.181** |         –          |
> | Linear_Projection |   0.341   |   0.369   |    0.381    |   0.186   |   $(d \times m)$   |
> | Gating_Projection |   0.343   |   0.354   |    0.386    |   0.184   | $(d^{2} \times m)$ |
> |   Concat_Linear   |   0.344   |   0.350   |    0.392    |   0.182   | $(d(d+1)\times m)$ |
>
> *Note: $m$ represents the number of Transformer encoders layers.
>
> # W3:
> **Response:** Thank you for this insightful comment.
>
>    ​	TRA's output—after MHSA, FFN, residual connections, and normalization—integrates weighted trends but does **not fully restore the original scale** for multi-step forecasting (Section 3.2). The Projection Layer (Eq. 9: $\hat{Y}^i = (P^i + \mu ^i) W_o + b_o$) re-incorporates $\mu$ as a calibration step, aligning predictions with input trends and preventing under-emphasis. Ablation studies confirm that removing the 2nd $\mu$ degrades performance, showing that the 1st addition alone is insufficient. The averaged results across all prediction lengths are shown below.
>
> |         | PMDformer |           | w/o the 2nd $\mu$ |       |
> | :-----: | :-------: | :-------: | :---------------: | :---: |
> |         |    MSE    |    MAE    |        MSE        |  MAE  |
> |  ETTh2  | **0.337** | **0.382** |       0.348       | 0.387 |
> |  ETTm1  | **0.346** | **0.371** |       0.349       | 0.372 |
> | Traffic | **0.378** | **0.234** |       0.393       | 0.258 |
> |  Solar  | **0.181** | **0.211** |       0.187       | 0.213 |

---

### Official Review · Reviewer_jQS8 · 2025-10-31

**Soundness:** 2
**Presentation:** 2
**Contribution:** 2
**Rating:** 2
**Confidence:** 4

**Summary:**

The paper proposes PMDformer, a Transformer-based model for long-term time series forecasting that preserves cross-patch and cross-variable shape similarities despite scale differences. It introduces three components: Patch-Mean Decoupling (PMD) to separate trend from residual shape by subtracting each patch’s mean; Proximal Variable Attention (PVA) to emphasize recent, relevant cross-variable relationships and mitigate overfitting to outdated correlations; and Trend Restoration Attention (TRA) to reintegrate global trend into the attention mechanism without distorting shape. The authors claim PMDformer achieves superior stability and accuracy over state-of-the-art baselines across multiple LTSF benchmarks.

**Strengths:**

The paper is well written and easy to follow.

The proposed method demonstrates superior performance compared to several existing baselines.

**Weaknesses:**

The paper’s motivation is unclear: there is no well-defined objective or problem statement, and the connection between the proposed method and the problem it aims to solve is not clearly articulated.

There is little theoretical or empirical justification for the design of the proposed PMDFormer; the choice of each component appears arbitrary and based on empirical intuition, with no systematic evaluation.

Efficiency is not discussed: the paper lacks a complexity analysis (parameter count, runtime, and memory usage) relative to the baselines.

The LLM usage statement is not in the original manuscript.

**Questions:**

please see weakness

---

> ### Author Response · Authors · 2025-11-21
> **Responses to Reviewer jQS8 (Part 1/2)**
>
> # W1:
> **Response:**
>
> (1). Our motivation is clear. The problem statement may have been misunderstood due to some of our wording, so we have revised a small portion to make it clearer. Our motivation and problem statement are as follows.
>
> ​	The paper now explicitly states two core problems in existing patch-based Transformers for long-term time series forecasting:
>
> ​	**Shape Similarity**: Existing methods fail to reliably capture shape similarity across patches and variables. This is because non-stationary scale differences in time series data distort dot-product attention logits, making them unable to reflect true shape similarity. Current patch-normalization techniques further aggravate this issue by dividing by standard deviation, which destroys the intrinsic shape structure (see revised **paragraphs 3 and 5** in the Introduction).
>
> ​	**Cross-variable Relationships**: When modeling cross-variable relationships, existing approaches (e.g., ModernTCN, Crossformer) compute interactions over the entire historical window. However, cross-variable relationships are highly non-stationary and evolve over time, so early correlations introduce significant noise and redundancy that degrade forecasting performance (see revised **paragraph 4** in the Introduction).
>
> (2). The connection between the proposed method and the problem is clearly presented in Introduction and Section Proposed Method.
>
> ​	Patch-Mean Decoupling (PMD) + Trend Restoration Attention (TRA) solve the **first** problem. PMD recenters each patch to zero mean and explicitly decouples the long-range trend, enabling the $Q$/$K$ pathway to focus **exclusively on shape similarity** without scale interference. The decoupled trend is then restored only in the Value pathway via TRA, preserving shape-driven attention scores while retaining global trend information for long-range modeling (see paragraphs 3 and 5 in the Introduction, full details in Section 3.2).
>
> ​	Proximal Variable Attention (PVA) solves the **second** problem by restricting cross-variable interaction to **the most recent patch**. This method **prevents outdated historical correlations** from contaminating the shape-focused attention mechanism, which may be the most relevant to the future sequence. (see paragraph 4 in the Introduction, full details in Section 3.2).
>
> # W2:
> **Response:**
>
> ​	The design of PMDFormer is systematically motivated by two specific failure modes of existing patch-based Transformers identified in **problem statement in Weakness 1**: the interference of scale differences with shape similarity capture and the ineffective modeling of cross-variable relationships. All designs revolve around these two issues. The main module include Patch-Mean Decoupling (PMD)，Trend Restoration Attention (TRA) and Proximal Variable Attention (PVA). Each module is designed to address the issue, and each is supported by **targeted ablation studies** that demonstrate the empirical justification.
>
> ​	(i). **PMD**: This module is designed to adress the **first** problem. PMD recenters each patch to zero mean and explicitly decouples the long-range trend, enabling the $Q$/$K$ pathway to focus **exclusively on shape similarity** without scale interference. This motivation is stated in Paragraph 3 of the Introduction and detailed in Section 3.2. Its effectiveness is systematically validated in **Table 3**, which compares PMD against alternative patch-wise normalization strategies.
>
> ​	(ii). **TRA**: This module leverages the trend decoupled by PMD to effectively model long-range dependencies. TRA adds the decoupled trend only to the Value pathway,  so attention remains purely shape-based while output tokens **carry global trend information**. The motivation appears in Paragraph 5 of the Introduction and Section 3.2, with ablation results in **Table 4** and further evidence in the response to Reviewer 1ngW Question 2.
>
> ​	(iii). **PVA**: This module captures **temporally relevant cross-variable relationships** by restricting attention to the most recent patch, **preventing historical noise** from contaminating the shape-focused attention and **reducing redundancy**. This design is motivated in Paragraph 3 of the Introduction and Section 3.2, with comprehensive ablation studies in Table 4, Section 4.3, and the response to Reviewer rABD Question 2.
>
> ​	Thus, no component is arbitrary. Each is purposefully designed to resolve the identified failure modes and is empirically justified through targeted experiments.
>
>
> # W3:
> **Response**:
> Thank you for pointing out the missing efficiency discussion. We have added a detailed complexity analysis, including FLOPs, parameter count, memory usage, training time, and inference time for both our method and leading baselines. The results are shown in Appendix A.1 and the response to Reviewer 1ngW Weakness 2.

---

> ### Author Response · Authors · 2025-11-21
> **More Responses to Reviewer jQS8 (Part 2/2)**
>
> # W4:
> **Response:**  Thank you for pointing this out. In the revised manuscript, we have added a clear LLM usage statement in the appendix (**Appendix B**), explicitly detailing that large language models were moderatedly used in the writing of this work, in full compliance with the ICLR 2026 guidelines.

---

> > ### Comment · Reviewer_jQS8 · 2025-11-27
> >
> > Thank the author for the responses. I have carefully read all the reviews and the authors' feedback to other reviewers. I still have concerns about the contribution and novelty of this work. In particular, the proposed method appears compositional, and I find that it is still not sufficiently justified from either a theoretical or an empirical perspective. I am open to other reviewers' comments regarding my opinion, but I will keep my current rating for now.
> >
> > I would like to see a concrete example (maybe a simple, extreme synthetic example), where existing methods fail but PMD succeeds. Alternatively, case studies that help explain why PMD works would strengthen the paper. For instance, in the ablation studies, PMD outperforms other methods on some datasets but shows indistinguishable performance on others. It would be helpful to understand whether certain inherent properties of the datasets account for these differences.

---

> > > ### Author Response · Authors · 2025-12-04
> > > **Response to Reviewer jQS8 (Part 1/2)**
> > >
> > > Thank you for your careful reading.
> > >
> > > ​	(1). `Contribution and Novelty && Compositional Methods`
> > >
> > > ​	**Contribution**:
> > >
> > > ​	This paper tackles a **specific and common issue** in highly non-stationary time series: **scale differences of patches often obscure true shape similarity**. As shown in **Figure 1** of our original paper, this “scale bias” can cause dissimilar shapes to receive higher attention weights than truly similar ones. This problem degrades both shape similarity learning and cross-variable modeling.
> > >
> > > ​	Our method tackles this with a simple yet effective redesign for Transformer architecture. The PMD and TRA modules let attention **focus on pure shapes** while restoring long-range trends for accurate forecasts. PVA further captures variable relationships and reduces redundancy. Thus, our contributions target key challenges in patch-based Transformer models for long-term prediction.
> > >
> > > ​	**Novelty**: Unlike RevIN (which normalizes the full series) or DLinear's decomposition method, PMD is the **first** to decouple scales for modeling shape similarity and we redesign the attention mechanism to reinject decoupled scale information to predict future trend. This method enables to model long-term trends without disrupting shape dependencies. The key differences are outlined below.
> > >
> > > |          Method           |                          PMD (Ours)                          |                            RevIN                             |                 DLinear / Autoformer                  |
> > > | :-----------------------: | :----------------------------------------------------------: | :----------------------------------------------------------: | :---------------------------------------------------: |
> > > |         **Scope**         |                          Per-patch                           |                          Per-series                          |                      Per-series                       |
> > > |         **Goal**          | Remove scale bias in $Q$/$K$ attention to focus on shape similarity (per-patch) | Improve generalization by addressing distribution shift (per-series) |   Separate slow-varying trend from other components   |
> > > |    **Core Operation**     |                     $P_j^i - {\mu}_j^i$                      |                   $\frac{X - \mu}{\sigma}$                   |          $X_{Seasonal} = X - \text{Pool}(X)$          |
> > > | **Relation to Attention** |             scale addition to the Value pathway              |    A pre/post-processing module independent of attention     | A pre/post-processing module independent of attention |
> > >
> > > ​	**Compositional**: While our modules build on Transformer basics, they form a **cohesive system designed for patch-based LTSF challenges**. PMD separates the residual shapes from scales, TRA restores the long-range trend for stability, and PVA focuses on recent variable shapes to avoid overfitting. These modules effectively capture shape similarities across temporal patches and variables, while maintaining long-term modeling. Ablations (Tables 3–4) also confirm their synergy, showing consistent gains on non-stationary datasets.
> > >
> > > ​	(2). `Synthetic Example’ to validate the effectiveness of PMDformer`
> > >
> > > ​	To validate the effectiveness of PMDformer, we built a synthetic long-term forecasting dataset consisting of alternating patches of two distinctive shapes: a sharp **pulse wave** and a slow **sine wave**. To simulate strong non-stationarity, we applied drastically **different mean levels** on the two shapes and added moderate Gaussian noise. We compared a standard patch-based Transformer (like PatchTST, denoted 'w/o PMD') against the same architecture equipped with our patch-mean decoupling (PMD) and trend restoration attention (TRA) modules ('w/ PMD').
> > >
> > > ​	As shown in **Figure 6** (**Appendix A.2** in the revised manuscript), the 'w/o PMD' model suffers from patch scale differences: its **predictions collapse** into almost flat smooth curves and completely **fail to recover the pulse pattern**. In contrast, the 'w/ PMD' model accurately reproduces the alternating pulse–sine sequence with **fitted shapes and trend**. This attributes to that PMD removes scale interference so that attention can focus on true shape similarity, while TRA restores the long-range trend information necessary for precise trend forecasting. This experiment highlights the necessity and effectiveness of our method.

---

> > > > ### Author Response · Authors · 2025-12-04
> > > > **Response to Reviewer jQS8 (Part 2/2)**
> > > >
> > > > ​	(3). `Inherent Properties of the Datasets`
> > > >
> > > > ​	To explain ablation differences (e.g., strong gains on ETTh2, Traffic, Solar; minor on ETTm1), we quantified **non-stationarity** (via KPSS test on sliding windows, rejecting stationarity at p<0.05) and **cross-variable correlation** (from TFB [1] leaderboard).
> > > >
> > > > ​	Quantity Analysis: We quantitatively analyzed two key properties of all benchmarks: degree of non-stationarity and strength of cross-variable correlation. Non-stationarity was measured with the KPSS test. For each dataset, we split the series into overlapping windows of length 720 (stride 360), computed the KPSS statistic per window, and reported both the average statistic (higher means more non-stationary) and the percentage of windows rejecting stationarity at p < 0.05. Cross-variable correlation strength was taken from the official TFB [1] leaderboard. The results are shown as follows.
> > > >
> > > > | Dataset | KPSS statistic | Non_stationary_Rate (%) | Num_Var | Var_Correlation |
> > > > | :-----: | :------------: | :---------------------: | :-----: | :-------------: |
> > > > |  ETTh1  |      0.87      |          58.05          |    7    |      0.63       |
> > > > |  ETTh2  |    **1.13**    |        **75.30**        |    7    |      0.51       |
> > > > |  ETTm1  |      0.60      |          38.91          |    7    |      0.61       |
> > > > |  ETTm2  |      0.85      |          56.32          |    7    |      0.50       |
> > > > | Weather |      0.92      |          55.35          |   21    |      0.69       |
> > > > | Traffic |      0.13      |          4.56           |   862   |    **0.81**     |
> > > > |   ECL   |      0.36      |          22.38          |   321   |      0.80       |
> > > > |  Solar  |      0.13      |          3.46           |   137   |      0.79       |
> > > >
> > > > ​	The results align with our ablation studies (Table 3-4 in the original paper). PMDformer achieves the **largest gains on ETTh2**, which shows the **highest non-stationarity** among all datasets. Though Traffic and Solar are highly stationary (low KPSS), they contain hundreds of variables with strong correlations yet very different scales. **Scale differences across variables create similar attention bias** as temporal non-stationarity, so PMDformer still provides clear benefits. In contrast, ETTm1 and Weather exhibit both **lower non-stationarity and lower variables correlation**, leaving little room for gains, which explains the smaller improvements.
> > > >
> > > > [1] X. Qiu, J. Hu, L. Zhou, X. Wu, J. Du, B. Zhang, C. Guo, A. Zhou, C. S. Jensen, Z. Sheng, B. Yang, Tfb: Towards comprehensive and fair benchmarking of time series forecasting methods, Proc. VLDB Endow. (2024).

---

### Meta-Review · Area_Chair_1v9t · 2026-01-06

**Summary:**

The reviews can be summarized as the following concerns:
1. Asking for detailed efficiency analyses.
2. Asking for providing the design priciples of several key modules.
3. Querying about the motivations of the PMD (i.e., shape similarities, scale bias).
4. Requiring to verify the effectiveness of the different modules, or more detailed ablations.
5. Considerations regarding novelty and robustness, e.g., comparison with RevIN, and the stability of the segmentation schema.
6. Inquring the selection of hyperparameters.

Among these concerns, doubts regarding motivation, novelty, and the rigor of experiments are the most critical aspects. In my opinion, the authors have made efforts and properly addressed these issues during the rebuttal phase, including:
1. Adding sufficient experiments to demonstrate the designs and efficiency.
2. Providing more theoretical and horizontal comparative analyses to prove the novelty.
3. Carefully updating the draft to update the corresponding revisions.

Considering that most (3 out of 4) reviewers hold a positive rating (i.e., 6) and the efforts made by the authors during the rebuttal period, I decide to recommend this paper as Accept (poster).

**Reviewer Concerns:**

I think most concerns are addressed by the rebuttal, there exist no outstanding ones.

**Reviewer Scores:**

Reviewer 1ngW thinked the concerns have been solve and choosed to maintain the positive attitude.

For Reviewer pzqK and rABD, I think their concerns are also well responded.

For reviewer jQS8, I think his/her reviews are not specific enough, and he/she may have given a subjective low score due to research
preference issues. I'm not quite sure what measures Reviewer jQS8 will take after participating in the discussion. Based solely on the rebuttal provided by the authors, i think the relevant issues in the draft have been resolved fairly well.

---

### Decision · Program_Chairs · 2026-01-26

Accept (Poster)